# The Necessity of a Global Binding Framework for Sustainable Management of Chemicals and Materials—Interactions with Climate and Biodiversity

Klaus Günter Steinhäuser [1,*], Arnim Von Gleich [2] , Markus Große Ophoff [3] and Wolfgang Körner [4]

1  Derfflingerstr. 14, 12249 Berlin, Germany
2  Department Technological Design and Development, Faculty Production Engineering, University of Bremen, Bibliothekstr. 1, 28359 Bremen, Germany; avgleich@aol.com
3  Ameldungstr. 14, 49082 Osnabrück, Germany; m.grosse-ophoff@hs-osnabrueck.de
4  Institute of Materials Resource Management, University of Augsburg, Universitätsstr. 2, 86159 Augsburg, Germany; lehmannkoerner@web.de
*  Correspondence: klaus-g.steinhaeuser@posteo.de; Tel.: +49-172-7607739

**Abstract:** Sustainable chemicals and materials management deals with both the risks and the opportunities of chemicals and products. It is not only focused on hazards and risks of chemicals for human health and the environment but also includes the management of material flows from extraction of raw materials up to waste. It becomes apparent meanwhile that the ever-growing material streams endanger the Earth system. According to a recent publication of Persson et al., the planetary boundaries for chemicals and plastics have already been exceeded. Therefore, sustainable chemicals and materials management must become a third pillar of international sustainability policy. For climate change and biodiversity, binding international agreements already exist. Accordingly, a global chemicals and materials framework convention integrating the current fragmented and non-binding approaches is needed. The impacts of chemicals and materials are closely related to climate change. About one third of greenhouse gas (GHG) emissions are linked to the production of chemicals, materials and products and the growing global transport of goods. Most of it is assigned to the energy demand of production and transport. GHG emissions must be reduced by an expansion of the circular economy, i.e., the use of secondary instead of primary raw materials. The chemical industry is obliged to change its feedstock since chemicals based on mineral oil and natural gas are not sustainable. Climate change in turn has consequences for the fate and effects of substances in the environment. Rising temperature implies higher vapor pressure and may enhance the release of toxicants into the atmosphere. Organisms that are already stressed may react more sensitively when exposed to toxic chemicals. The increasing frequency of extreme weather events may re-mobilize contaminants in river sediments. Increasing chemical and material load also threatens biodiversity, e.g., by the release of toxic chemicals into air, water and soil up to high amounts of waste. Fertilizers and pesticides are damaging the biocoenoses in agrarian landscapes. In order to overcome these fatal developments, sustainable management of chemicals and materials is urgently needed. This includes safe and sustainable chemicals, sustainable chemical production and sustainable materials flow management. All these three sustainability strategies are crucial and complement each other: efficiency, consistency and sufficiency. This obligates drastic changes not only of the quantities of material streams but also of the qualities of chemicals and materials in use. A significant reduction in production volumes is necessary, aiming not only to return to a safe operating space with respect to the planetary boundary for chemicals, plastics and waste but also in order to achieve goals regarding climate and biodiversity.

**Keywords:** planetary boundaries; chemicals management; materials flow management; climate change; biological diversity; sustainable chemistry; efficiency; consistency; sufficiency

## 1. Introduction

Humans have become one of the most important factors influencing biological, geological and atmospheric processes on Earth. Paul Crutzen created the term Anthropocene to reflect this new geological period [1]. Global problems and risks need global governance. Regarding biodiversity and climate change, international boards and framework conventions already exist. There is currently no corresponding global board and agreement for the chemicals and materials sector.

Regarding biodiversity, there is the Convention on Biological Diversity supplemented by the Cartagena and the Nagoya Protocol. Targets are the conservation of biological diversity, the sustainable use of its components, and the fair and equitable sharing of benefits arising from genetic resources. The Intergovernmental Platform on Biodiversity and Ecosystem Services (IPBES) delivers scientific reviews. Regarding Climate Change there is the United Nations Framework Convention on Climate Change supplemented by the Kyoto Protocol and the Paris Agreement for Climate Protection in order to combat dangerous human interference with the climate system and to stabilize greenhouse gas concentrations in the atmosphere. Scientific advice delivers the Intergovernmental Panel on Climate Change.

In this paper we provide figures and facts in order to exemplify why, as a third pillar of global governance, a legally binding international chemicals and materials framework agreement is urgently required. Additionally, we provide approaches that contribute to solving some of the most pressing problems by reduction in material streams and higher efficiency, by reducing consumption (sufficiency), by substitution of hazardous substances, by synthesis based on green hydrogen (Power-to-X), by recycling and not least by designing substances as benign from the start.

Main reasons for concern in view of substance and materials streams are first the uncontrolled spread of persistent, mobile and in part bioaccumulative and toxic substances and materials, including plastics and second the ever-increasing exploitation of natural resources (biologic and mineral) with all their requirements of fossil energy as well as emissions and waste streams. A third focus lies on the interaction between the global chemicals and materials streams on the one hand and climate change and loss of biodiversity on the other. An integrated view of this acute situation is provided by the approach of planetary boundaries, which have already been crossed and are in danger of being crossed in the near future, respectively.

Such a legally binding international chemicals and materials framework agreement must clearly contain defined qualitative specifications and quantitative reduction targets for the consumption of chemicals and resources as well as a strategy for sustainable detoxification of the environment, efficiency targets and requirements for the management of material cycles and waste treatment. The SAICM (Strategic Approach to International Chemicals Management) process may be an initial point. In this context, it is also important to create an independent intergovernmental body as a science–policy interface—analogous to the IPCC and the IPBES. Numerous prominent scientists have drawn attention to this need [2] and made clear that it is important to scientifically undergird a sustainable chemicals and materials policy [3]. Initiated by Switzerland and other countries, UNEA 5.2 adopted a resolution in February 2022 to establish an intergovernmental panel [4]. The International Panel on Chemical Pollution (IPCP) could possibly be a building block of such an advisory body [5].

## 2. Chemicals and Materials Affect the Stability of the Earth System

Sustainable chemicals and materials management extends far beyond what is traditionally viewed as sound management of chemicals. In this publication, the term "substance" is understood in a very comprehensive way. Whereas conventional chemicals management is focused on hazardous properties of substances and risks for human health and the environment arising from them, the overarching concept of chemicals and materials management also includes and connects the management of raw materials, resources, products and

wastes. The ever-growing material and waste streams also pose a threat to environment and human health. Enhancement of material streams implies that exposure of humans and the environment to the chemical ingredients of the materials increases as well.

Following the framework conventions on climate change and loss of biodiversity, the chemicals and materials management too must be based on the principles of precaution and sustainability. The precautionary principle requires action to be taken whenever there are comprehensible reasons for concern, even when no conclusive evidence of a causal relationship is called for [6], whereas sustainability means meeting the needs of today's generation without impairing the needs of future generations [7]. Sustainability and precautionary action largely cover the same issues in their orientation, with sustainability focusing more on global and long-term effects, and precaution taking greater account of knowledge deficits in view of impending irreversible effects. Without suitable environmental precautions, sustainable development is not possible.

In 2015, the General Assembly of the United Nations adopted seventeen goals for sustainable development that are to be achieved by 2030 [8]. These Sustainable Development Goals (SDGs) include a number of environmental and health-related goals such as clean drinking water for all people, the protection of terrestrial and marine ecosystems and also the preservation of health and the environment through fewer dangerous chemicals and less pollution of water, soil and air. Several of these goals specifically refer to the impact of substances. In particular, SDG No 12 ("Sustainable production and consumption") and the targets 12.2, 12.4 and 12.5 require materials policy action.

Chemicals and materials management is an international challenge. Several international treaties such as the Rotterdam, Stockholm, Basel and Minamata Conventions and the Montreal Protocol aim at minimizing risks arising from chemicals and waste. Negotiations on a global treaty on plastic pollution will be launched in February 2022 at UNEA 5 [9]. The Strategic Approach to International Chemicals Management (SAICM) is an essential initiative for the sound management of chemicals and waste and aims to achieve a common understanding of the global challenges to manage chemicals and waste [10]. However, currently the various measures and forums are fragmented and rarely coordinated.

In 2009, Rockström et al. [11] attracted worldwide attention by the publication of "A safe operating space for humanity." The authors presented a scientific approach with the purpose of describing the stability of our planet and defining the planetary boundaries of what Earth can withstand. According to Steffen et al.'s "Planetary boundaries: Guiding human development on a changing planet", human activities have reached a level that could seriously disturb the stability of the systems that hold Earth in its current state [12]. They describe nine processes that are decisive for the stability of the Earth system (Figure 1).

One of these processes is 'novel entities' which describe the burden on the Earth system by anthropogenic substances as well as modified life forms such as products of synthetic biology. In 2022, an international team of researchers [13] assessed the impact on Earth system stability of the cocktail of synthetic chemicals, plastics and other "novel entities" flooding the environment. The researchers assessed the suitability of a set of control variables such as quantities of production, quantities of release to the environment and share of hazardous chemicals which exhibit characteristics such as persistence and mobility. They concluded that, despite data limitations, Earth system processes are increasingly disturbed, and the Earth system is at risk. Humanity has crossed the planetary boundary for novel entities.

Since 1950, chemical production has increased 50-fold. By 2050, that amount is expected to triple again [14]. There are an estimated 350,000 different types of chemicals produced on the global market. These include plastics, pesticides, industrial chemicals, chemicals in consumer products, antibiotics and other pharmaceuticals. These are entirely new substances that have been created by human activities and whose effects on the Earth system are largely unknown. Significant quantities of these novel substances enter the environment every day. The authors call for urgent action to reduce the production and release of these pollutants. Inter alia they call for caps on production of chemicals and plastics.

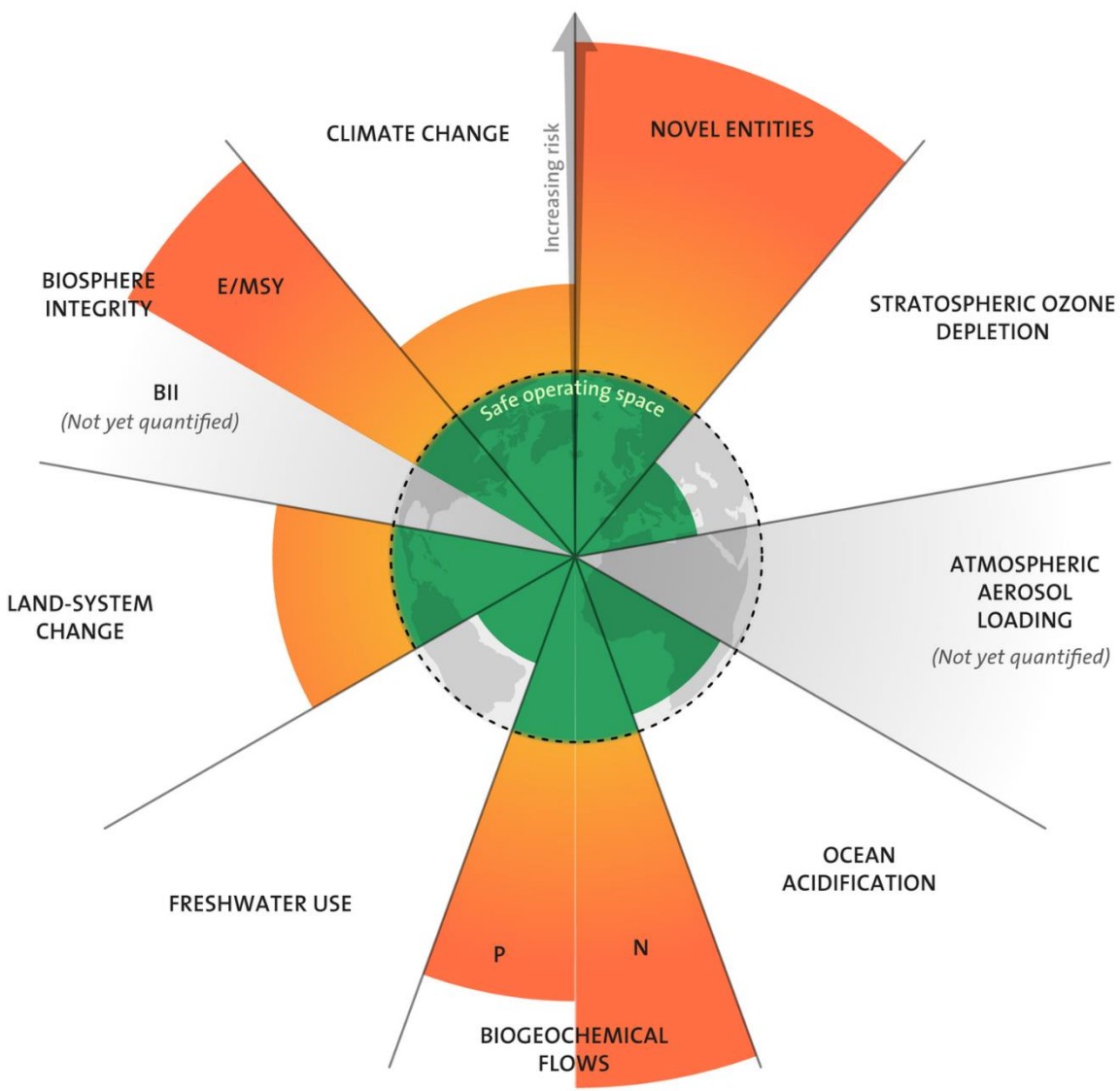

**Figure 1.** Planetary Boundaries: processes which can disturb the stability of the Earth; green zone: safe operating space; red zone: planetary boundaries are exceeded. Designed by Azote for the Stockholm Resilience Centre, based on an analysis in Persson et al. 2022 [13] and Steffen et al. 2015 [12], BII: functional diversity (Biodiversity Intactness Index), E/MSY: genetic diversity (extinctions per million species-years).

The chemical pollution of our planet also influences other processes such as atmospheric aerosols, biogeochemical flows of nitrogen and phosphorus and stratospheric ozone depletion [12]. A well-known and quantifiable global boundary relates to climate change, based on continuous high release of $CO_2$ $CH_4$, $N_2O$ and other greenhouse gases and resulting in losses of biodiversity. The rapid increase in the use of non-renewable resources [15] and materials pollution shows that there is a clear need for concerted action by the international community. The "great acceleration" of numerous ecological and socio-economic parameters through human activities [16] is apparently linked also to the production and use of chemicals and materials.

Scientists have formulated three criteria that "novel entities" in the form of synthetic chemicals must fulfil in order to have a global impact: (i) they are persistent (stable in the environment for extended periods of time), (ii) they are mobile over long distances such as climatic zones or continents and correspondingly widespread, and (iii) they are able to affect important processes of the Earth system or its subsystems [7]. In any case, the release of highly persistent and mobile chemicals into the environment is generally

problematic and requires precautionary action. Their irreversible impact on ecosystems and human health extends over long periods of time. They can spread from the point of release through wind or water and accumulate in organisms of terrestrial and aquatic food webs. Therefore, persistence is a key feature that significantly contributes to high human and environmental exposure to chemicals [17,18]. Conventional substance evaluation to date is normally based on comparing effect thresholds and predicted exposure. If the predicted or measured exposure concentrations/doses are higher than the effect threshold, a risk is identified which needs to be reduced. This well-proven approach ignores that exposure and effects are temporally and spatially decoupled regarding persistent and mobile substances. As they spread and accumulate in the environment, effects can be transferred to somewhere else. Apart from persistence, the mobility of substances is a very problematic feature which has been underestimated for a long time. This concerns mobility in the water cycle and also long-range atmospheric transport. Criteria for the identification of persistent, mobile and toxic (PMT) chemicals have been developed [19] and will be implemented in EU chemicals management. If adverse effects are observed at a later date, the substances cannot be removed again from the environment. Even without a known (as yet undetected) negative effect, persistent chemicals thus have a high hazard potential. They can stay in the environment for a long time, spread widely, accumulate in certain places and lead to completely unexpected interactions with various substances and organisms. This is impressively demonstrated by chlorofluorocarbons (CFCs) and microplastics which are non-toxic but pose severe environmental problems. Per- and polyfluorinated alkyl substances (PFAS), used in numerous applications, are additional examples of extraordinarily persistent substances. They remain resistant in the environment for years to decades and are therefore called "forever chemicals" [20]. Some PFAS (e.g., PFOS (perfluorooctanesulfonic acid) and PFOA (perfluorooctanoic acid)) are already restricted by the Stockholm Convention and the EU but are now substituted with other PFAS which have in common their extreme persistence. An assessment and restriction of the whole group of PFAS is necessary and envisaged by the EU [21], aiming at a rapid and total phase-out of this group of substances.

In 2020, the EU Commission published its "Chemicals Strategy for Sustainability" in order to further develop the REACH regulations. It is also aiming at so-called non-toxic material cycles [21]. Some elements of this enhanced strategy like the principle "one substance—one assessment", the introduction of a mixture assessment factor, the facilitation of group regulations and the trend towards inherently safe chemicals will be important steps towards sustainable chemicals management.

## 3. Production and Use of Substances Interact with Climate Change

Climate protection is currently the dominant issue in environmental policy. The dramatic consequences of global warming can be felt worldwide. In order to limit global warming to 1.5 °C compared with the pre-industrial age, in the 2015 Paris Agreement the signatory states committed themselves to reducing their greenhouse gas (GHG) emissions accordingly. Figure 2 shows that, in 2016, 29.4% of total global greenhouse gas emissions (48.9 Gt $CO_2$eq in 2018) were attributable to the production of substances (energy use and other emissions). Additionally, the road transport sector (road, shipping, aviation, railway) emitted 11.9% [22]. According to the ITF report 2021, more than 40% of traffic emissions come from freight transport [23].

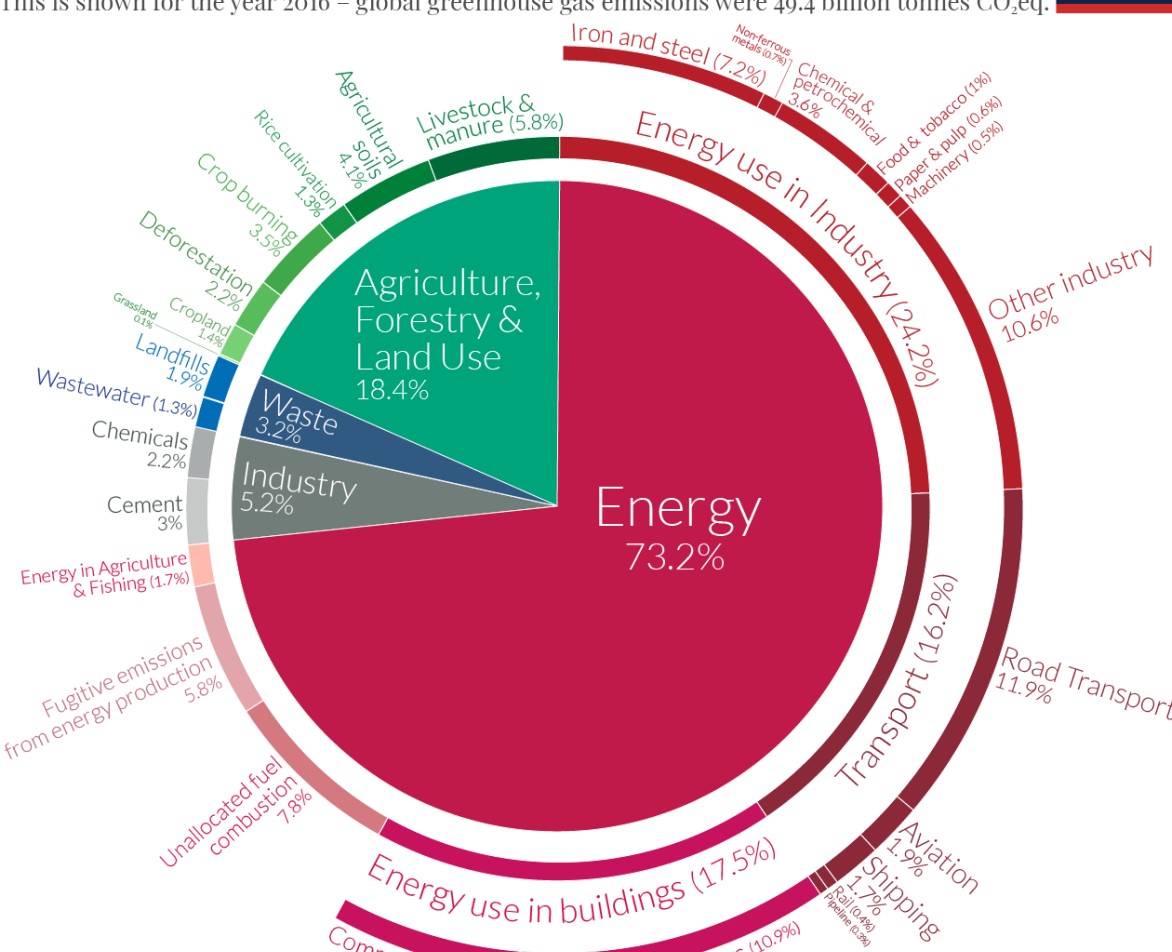

**Figure 2.** Global greenhouse gas emissions by sector in 2016 according to Ritchie [22].

### 3.1. Expenditure of Energy for Production and Use of Substances

The production of chemicals and materials requires energy, beginning with the extraction and refining of raw materials and extending to the synthesis of chemicals and the processing of products. Across the globe, energy is still obtained mainly by burning fossil as opposed to renewable fuels such as coal, oil, and natural gas. Thus, a shift of energy production away from fossil to renewable resources would also help to reduce GHG emissions caused by the production of chemicals and materials. Chemicals, materials and products differ in the specific energy expenditure required for their fabrication. Manufacturing methods and the equipment of production plants lead to different levels of GHG emissions. The "cumulative energy demand" (CED) [24], which covers the entire life cycle of a substance, and the sum of the associated GHG emissions are indicators of the climate impact of particular products.

The final energy productivity in Germany today is 60% higher than in 1990. This means that the energy consumption was efficiently decoupled from economic growth (Gross domestic product GDP) (Figure 3). However, absolute energy consumption has not decreased as a result, but has virtually remained at the same level in Germany [25] and is even continuing to increase worldwide [26]. The reason is the so-called "rebound effect":

specific savings are compensated by economic growth and changes in use. Consequently, if one really wants to reduce energy consumption and the associated GHG emissions involved in the manufacturing of substances, it is necessary to avoid, reduce and slow down the flow of materials. Increasing the longevity of products and ease of their repairing, as well as an enhancement of reuse and recycling of products, is a challenge faced by manufacturers, politicians and consumers alike.

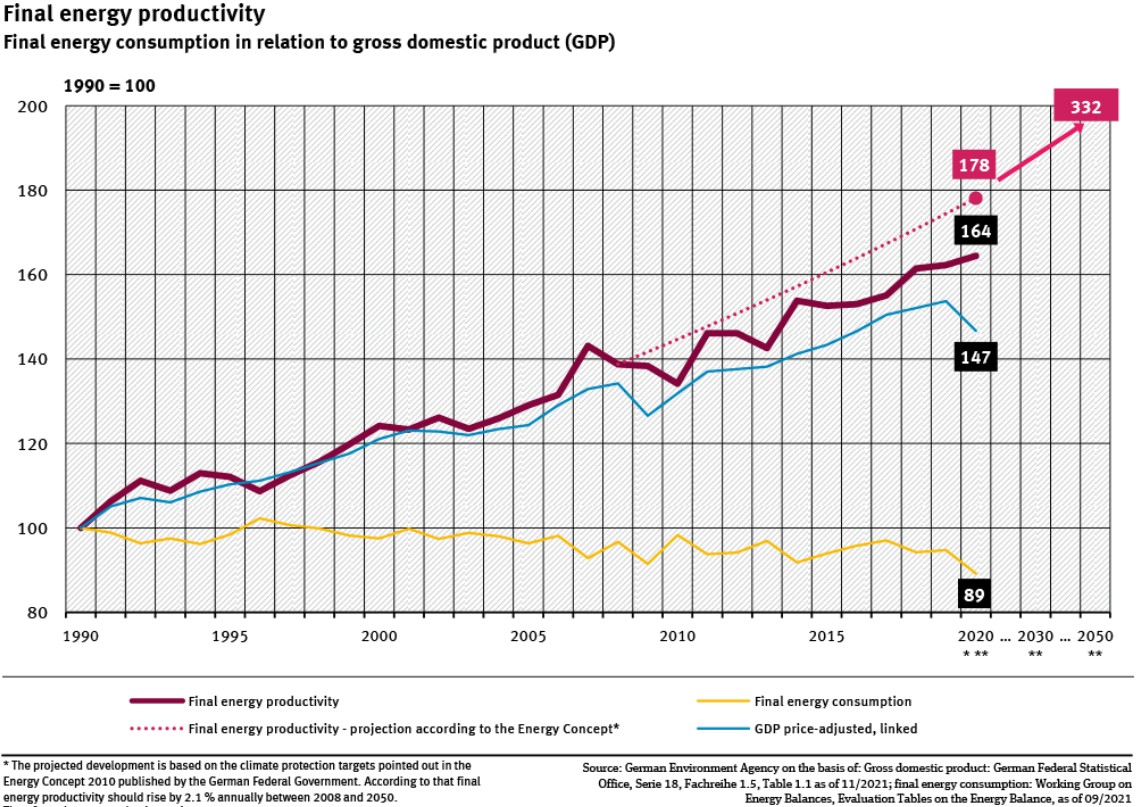

**Figure 3.** Final energy productivity 1990–2020 (Source Umweltbundesamt [25]).

*3.2. Energy-Intensive Production Processes*

There are some production processes where primary energy demand is extraordinarily high such as the synthesis of ammonia from nitrogen and hydrogen. Every year 150 Mt of ammonia ($NH_3$) is produced by the Haber–Bosch process all over the world, mainly for fertilizer. This involves emissions of approximately 300 Mt $CO_2$eq [27]. Incidentally, the production and the use of too much ammonia-related fertilizers are the main reason for the frequent breaches of the planetary boundaries for reactive nitrogen. Other examples are the production processes of cement and steel. In these cases, the GHG emissions are not only caused by the energy demand but also result from chemical reactions generating $CO_2$.

During the production of cement, carbon dioxide is emitted in the process of the decomposition of lime (calcium carbonate) to calcium oxide. These emissions are responsible for approximately two-thirds of carbon dioxide emissions in the course of the process, which according to the International Energy Agency (IEA) is responsible for 6.9% of worldwide $CO_2$ emissions [28]. Possible solutions to reduce these GHG emissions are the substitution of concrete in construction (e.g., timber), partial substitution of cement in concrete (e.g., by blast furnace slag from steel mills, rice husk ash) or reduction in the percentage of clinker in cement (e.g., ultra-lightweight concrete with a high air content). Coal or oil used for heating of the cement kilns can be substituted by waste (as already usual in many countries) [29–31], but far more GHG reduction can be achieved by carbon capture and use or by electrical heating from renewable power plants.

In the metal industry coal is used as a reducing agent in the smelting of many ores. The use of blast furnaces to produce raw iron, an intermediate product for steel, is a particularly high emitter of $CO_2$. This process is responsible for 8–9% of GHG emissions worldwide.

Iron can be produced without using coal as a reducing agent. There is increasing debate on using hydrogen instead of coal and thus potentially making this process greenhouse-gas-neutral [32]. A pilot plant for steel production with hydrogen is being built in Sweden [33]. In Germany, steel manufacturers are trying to replace some of the coal in the blast furnace with hydrogen. But most of the hydrogen is still obtained from natural gas and is therefore associated with GHG emissions ("gray" hydrogen). The production of regenerative "green" hydrogen (hydrogen which is produced by electrolysis of water with renewable electricity) is only just at the beginning. Due to the large quantities of regenerative energies required, complete conversion to "green" hydrogen will only be possible if at the same time the production of iron and steel is substantially reduced through systematic recycling and sufficiency strategies.

*3.3. Replacing Primary Raw Materials*

Without strict recovery and recycling of resources after use, the planet's resources will be exploited at ever greater speed. Consequently, in its action plan for a circular economy, the European Commission presents ways and means of transforming the current system of more or less linear flows with increasing quantities of waste into a system in which the reuse of products and the recycling of materials is favored [34]. The replacement of primary raw materials with those that have already been used once or several times (secondary raw materials) can, in addition to conserving resources, also significantly reduce total energy consumption in the manufacture of products. As a first approximation, this is shown by a comparison of the cumulative energy demand (CED) [24] for the production of the primary raw material with the CED for its recovery from products. The former can be found in studies about many important primary raw materials [35]; the latter is highly dependent on the product from which the raw material is recovered: the recycling of production waste is easier and requires less energy than the sorting out and recycling of mixed materials from used products. The entropy factor plays a decisive role here: this means that the greater the dissipation of the raw materials to be recovered from waste, the more exergy (exergy refers to the part of the total energy of a system that can do work when it is brought into thermodynamic equilibrium with its environment) and costs is required to recover it in a reasonably pure form. Approaching a recycling rate of 100%, the energy and material requirement for recovery increases tremendously [36]. However, it is not just the energy consumed that plays an important role. The cumulative resource demand (CRD) [37] describes the amount of materials that are required to produce a specific amount of the particular raw material. Secondary raw materials have the advantage that their ecological rucksack, that is to say the amount of unusable overburden, is usually less than that of primary products.

There are further obstacles to boost the recycling rate: the lack of transparency concerning the material composition of many products impedes a high-quality recycling if recyclers do not know the contamination with impurities and interfering tramp elements or even toxic chemicals. In addition, products sometimes contain hazardous sub stances that prevent recycling [38]. In other cases, especially regarding plastics, the secondary material is economically not competitive with the primary material. In these cases, economic steering instruments aiming at a privileged use of secondary materials should be taken into consideration.

The difficulties that arise with recycling can be illustrated with three examples:

The recycling of metals such as iron, copper and aluminum is widespread and well established as long as the scrap metals fulfil certain purity requirements. However, accompanying tramp elements that cannot be technically removed at present (such as copper in steel or aluminum) cause increasing problems when more and more metals are in the cycle and less primary raw material is available for dilution. In the worst case, the contaminated

material has to be removed from the cycle and disposed of [39]. Recycling is less common with rare metals (e.g., cobalt and lithium). Especially in the case of rare and strategically important metals, and as a contribution to resource conservation, it can make ecological, economical and also political sense to produce a secondary raw material requiring more energy for recycling than needed for the primary raw material. This can be the case if the extraction of raw materials in a third country is ethically and ecologically problematic and/or a country does not want to be totally dependent on a single exporter. It is expected that the demand for raw materials required for the generation and storage of energy from renewable sources will increase dramatically in the forthcoming years as a result of the expansion of future technologies. For example, an increase in annual consumption from 30 Mt to 110 Mt is expected for lithium in the period between 2013 and 2035 [40].

In the case of concrete, recycling can save up to 66% of the energy required for production, if entire walls are integrated in new buildings [41]. More important may be the aspect of resource conservation in the case of sand and gravel. Digging of sand and gravel destroys natural landscapes and biotopes. For concrete production, suitable sand already becomes a rare good resource. Consequently, recycled concrete and secondary aggregates should be used much more in building construction. Appropriate systems are available, e.g., from Germany [42], Denmark [43] and Brazil [44].

Plastics are much more problematic to recycle, not least because of their variety and countless additives. Since the 1950s, more than 8 Gt of plastics have been produced globally, with production doubling approximately every 20 years and amounting to more than 400 Mt in 2015 [45]. Plastics are highly persistent substances that are distributed in the environment in substantial quantities through abrasion, wear and tear or at the end of their lifecycle. In the EU in 2015, the GHG emissions for the manufacture, processing and disposal of plastics corresponded to an annual budget of 1781 Mt $CO_2$ equivalents [46].

More than one third of the plastic produced in Europe is used for packaging [40]. Many packaging films consist of multilayer systems that make single-substance recycling impossible. However, some sorts of plastic are recycled with great success from post-consumer waste, especially PET beverage bottles or HDPE [47]. The recycled bottles are even suitable for food and cosmetics [48]. A further benefit is that plastic bottles that remain in the recycling cycle do not end up in the environment and do not pollute soil or water by being broken down into pieces of microplastic over the course of centuries. The very stringent requirements for the purity of secondary plastics in contact with foodstuff are an additional problem. Recycled products often do not meet the requirements. Conservative, precautionary consumer protection then conflicts with the goals of circular economy [49]. For mixed plastics in post-consumer waste, it often makes no ecological sense to use much energy for separating, cleaning, pelletizing and re-extruding contaminated plastics. It may be reasonable to use a certain amount for manufacturing cheap products such as bollards or park benches ("downcycling") but the market niches for these objects are limited. Before such waste is used exclusively for energy in waste incineration plants or cement kilns, it can be advantageous to apply chemical recycling, such as through depolymerization, during which the monomers that make up the plastic are recovered, or by solvolysis such as the CreaSolv process, with which multilayer packaging materials can be processed [50]. This also enables to separate the persistent, bioaccumulative and toxic flame retardant Hexabromocyclododecane (HBCD) from polystyrene. So-called "raw material" recycling by pyrolysis or gasification is currently discussed widely. The products are synthesis gas and/or oils. These processes should be viewed critically because of the formation of hazardous by-products like tar. The oils and gases generated should not be used as fuels; direct incineration would be a better choice for unrecyclable polymers [51]. However, pyrolysis may be a means to close the carbon cycle and use its products as feedstock for the production of chemicals (see below) [52]. Overall, several procedures are currently being tested [53], but they still have to prove that they make economic and ecological sense.

In addition to the development and application of technical processes, the expansion of recycling also includes the establishment of transport and collection logistics, measures

to increase the acceptance of secondary raw materials in trade and among consumers and, above all, changes in product design. Products must be reusable, more durable, repairable and have a modular design so that high-quality recycling products can be made from them. Therefore, recyclability must be an essential evaluation criterion for products (see also [54]). In addition, regional value chains avoid long transport routes. An effective circular economy can be an effective means to reduce the consumption of energy and resources in the manufacture and use of products, even though nearly 100% material recycling remains an illusion for many groups of substances.

### 3.4. Production of Chemicals

Global chemical production is continuously increasing in terms of quantity, turnover and diversity [9]. In recent decades, production has roughly doubled every twelve years. A corresponding further increase is predicted. The chemical industry consumes around 10% of global energy demand for its processes. Mineral oil consumption in the chemicals sector is growing the fastest compared to other industries [55]. Though the chemicals industry is making considerable efforts to use energy efficiently, e.g., by combining production processes, and thus to improve its carbon footprint, in the long run; however, it cannot be sustainable to use fossil feedstock—above all mineral oil and also natural gas—as the dominant raw material base for production. Ultimately, substances that are made from fossil raw materials also contribute to greenhouse gas emissions because they are either biodegraded or incinerated at the end of their lifecycle. Coal, the formerly widely used source of raw materials, is no alternative because of its even greater $CO_2$ emissions per unit of energy compared to mineral oil and natural gas. The direct use of the low-exergy and inert carbon dioxide as a synthesis building block (Carbon Capture and Use, CCU) can be applied for some syntheses. For example, $CO_2$ may react with epoxides to yield organic carbonates [56] or lactones [57]. Artz et al. analyzed the potentials and the impacts of these types of $CO_2$ conversion [58]. In all cases you need energy, highly reactive reaction partners and appropriate catalysts to overcome the inertness of the $CO_2$ molecule. Therefore, the scope of the use of $CO_2$ as synthesis building block is rather limited. This leaves three options for obtaining carbon for the synthesis of organic substances which may be needed altogether to achieve a fossil-free chemical production: either from products generated by chemical recycling of polymers, from biological raw materials or synthetically from carbon dioxide and hydrogen.

Polymer waste which cannot be recovered by mechanical recycling can be treated by chemical recycling (e.g., solvolysis, pyrolysis, gasification) in order to recapture basic chemicals for chemical synthesis (see above) [52]. It has to be proven that these processes are ecologically reasonable.

Biological substances have the advantage that nature's synthesis processes, such as photosynthesis in plants and those occurring in microorganisms can be used. "White" biotechnology in closed systems is being used for the synthesis of an increasing number of chemicals. Algae cultures can also utilize organic residues and produce biomass in the form of proteins, fats and carbohydrates [59,60]. In addition, natural materials such as fiber plants and wood are often a sensible alternative to, for example, concrete as a building material or to plastics. When producing pulp from wood, large amounts of lignin are left over and used as a "biofuel" when producing pulp and paper [61]. However, there are a lot of research efforts to use lignin as a suitable raw material for the production of various chemicals, in particular aromatic compounds [62].

At the same time, the potential of a bioeconomy, that is, an economy in which biological materials and bioprocesses are used extensively, is limited. Only a fraction of the current demand for mineral oil as feedstock could be replaced by biological raw materials. That is because the production of biomass is mostly tied directly to available acreage. Arable land cannot be increased either in the countries of the Global North nor in the Global South. Competition for land in order to cultivate food-delivering plants implies a further

intensification of agriculture and forestry. Destruction of soils and natural areas would inevitably result [63].

Biomass can also be converted into basic chemical substances through hydrogenation (Fischer–Tropsch synthesis). In doing so, however, the valuable level of complexity of substances in natural products is destroyed. If possible, this complex molecular level should not be reduced through industrial processes, such as by biorefining natural materials to the level of ethene or simple hydrocarbon mixtures such as naphtha followed by resynthesizing new complex molecules combined with intensive energy use and numerous by-products. In general, it is more environmentally friendly to use waste biomass as feedstock rather than cultivated biomass.

The third alternative is the synthesis of basic chemical substances through the reaction of carbon dioxide ($CO_2$) with hydrogen ($H_2$) using energy. With such Power to X processes (PtX), not only methane but also synthetic fuels can be produced. Numerous smaller plants are already in operation [64,65]. Basically, this only helps to mitigate climate change if the energy used to produce hydrogen ("green" hydrogen) and the base chemicals synthetized from it is from renewable resources. Due to current low efficiencies, considerable amounts of energy are required for these processes, which a long time have not been taken into account in estimates of future energy demand. According to information from the chemicals industry in Germany, the consequent shift towards PtX in this sector would result in an increase in electricity demand by a factor of 11. Its GHG emissions would be reduced by 98% until 2050 [66]. When burning fossil fuels is no longer practiced, demand for $CO_2$ will have to be met by filtering it from the air [67,68]. Methods for "direct air capture" are currently being developed but are still energy- and cost-intensive.

It is becoming clear that a stabilization of the current production level will not suffice. A reduction in chemical production will be necessary to meet sustainability goals. Just another argument for reducing material flows not only through increase in efficiency, but also by material cycles (consistency) and by reducing consumption (sufficiency).

### 3.5. Transport of Substances and Products

Not only production and use of substances contribute to the rise of GHG emissions. Together, the rising volume of material flows and longer and more complex value chains are causing a rapid increase in (international) freight traffic. Both road (and rail) transport and shipping are major emitters of greenhouse gases that are still barely regulated. The greenhouse gas emissions from this area are increasing and amount to more than 3.0 Gt in 2020 [23]. At present, more than 80% of global trade volume is handled via shipping. Shipping contributes to global $CO_2$ emissions by around 2% [69].

### 4. Climate Change Impacts Fate and Effects of Substances

Climate change affects a lot of physical, chemical and biological processes on Earth: Higher temperatures, changing amounts of precipitation, shrinking ice cover, thawing of permafrost soils, changes in vegetation, ocean and air currents, all these factors also have an impact on the dispersion, exposure and in some cases effects of pollutants on people and environmental organisms. Numerous stress factors affect people and environmental organisms. They can increase the sensitivity to toxic substances and the likelihood of illness. Climate change is shifting the times of reproduction and ingestion of food, which can lead to reduced fitness of the organisms, e.g., of fish and zooplankton [70].

### 4.1. Larger Amounts of Toxic Air Pollutants

As temperatures rise, the tendency of solid and liquid chemicals to be vaporized increases significantly. Since the vapor pressure of a substance is temperature-dependent, the concentrations of air pollutants such as volatile organic compounds (VOC) will increase with rising temperatures. In addition to air temperature, the duration and intensity of solar radiation on materials and products can play an important role in the volatilization of chemicals. This has recently been demonstrated for the release of organic compounds,

including many aromatic hydrocarbons, from the asphalt of road surfaces in California: with moderate direct sunlight exposure on the asphalt, emissions increased significantly. The vaporized substances are oxidized by hydroxyl radicals to form semi- and non-volatile compounds, which aggregate and form aerosols. These secondary organic aerosols can be inhaled. Extrapolated for the region of the metropolis of Los Angeles, the annual emissions from asphalt in summer already exceed the primary particulate matter emissions from road traffic in the city [71].

In addition, VOC entering the lower troposphere in the presence of nitrogen oxides ("NOx": NO and $NO_2$) and intense solar radiation result in the formation of high ozone concentrations [72]. Ozone is a respirable gas that has an irritant effect on the respiratory tract. It triggers inflammation and leads to impairment of lung function.

Climate change also increases the frequency and duration of summer heat waves. The associated decrease of the frequency of rainfall events in summer can lead to higher concentrations of air pollutants, especially particulate matter, since rain-out and wash-out processes are effective mechanisms to remove aerosols from the air. As a result, in spite of the emission reductions in VOC and NOx that have already been achieved, increasing concentrations of air pollutants may be expected. In urban areas in particular, the temperature will rise significantly faster than the global mean [73]. In 2050, compared to 2020, it is expected that Berlin's summer will be on average up to 6 °C hotter; for Vienna, temperatures even more than 7.6 °C are predicted [74,75]. Experimental studies confirm combined effects of various air pollutants such as nitrogen oxides, inhalable particulate matter and ozone [76]. In urban areas in particular, people are already subjected to a high degree of stress by summer heat waves, can become dehydrated and as a result react with greater sensitivity to air pollutants. In August 2020—the second hottest month since weather records began to be kept in 1881—6% more people died in Germany than the average between 2016 and 2019 [77]. There is evidence that the combination of climate change and exposure to air pollutants has the potential for serious adverse effects on human health [78]. What applies to humans also applies to animals and plants: air pollutants affect metabolism and growth in flora and fauna.

### 4.2. Semi-Volatile Substances

Global warming is causing a greater transfer of semi-volatile chemicals into the atmosphere. Semi-volatile chemicals are substances that do not evaporate quickly but have such a high vapor pressure that they occur at least partially gaseous in the atmosphere and not only attached to fine particulate matter. This includes most of the persistent organic pollutants (POPs). Their atmospheric long-range transport and global distribution is intensified by global warming. Typical representatives are the highly fat-soluble polychlorinated biphenyls (PCBs). Although the production of PCBs was terminated during the 1980s all over the world, they are currently still being released from materials containing PCBs such as joint sealing compounds and paints [79]. The PCBs spreading around the globe through long-range atmospheric transport are thus able to contaminate land and water surfaces by wet and dry deposition, especially in arctic and subarctic regions. These global processes of atmospheric transport and deposition are called "global distillation" or the "grasshopper effect" [80]. PCBs and other POPs are ingested by organisms and bioaccumulate in animals at the top of the food webs in high concentrations [81]. Additionally, from higher altitudes in alpine regions, rather high PCB levels in wildlife are reported [82].

Substantial amounts of POPs have been deposited on glaciers and arctic soils over the decades. The melting of glaciers and the thawing of permafrost soils as a result of global warming are increasingly releasing these substances with melt and spring waters (as well as into the atmosphere) [83]. This leads to elevated POP concentrations in water bodies, soil, and finally in organisms [78]. On the top of the Mount Zugspitze in Germany, organochlorine pesticides, PCBs and other POPs have already been detected in the spring water of the tunnel system below the glacier (Schneeferner) [84].

Significant vaporization and atmospheric transport of semi-volatile pesticide ingredients from plant and soil surfaces after application has been demonstrated by the widespread entry of the herbicides pendimethalin and prosulfocarb into organically managed agricultural areas of the Schorfheide-Chorin biosphere reserve in northeastern Germany for several years [85]. In South Tyrol, too, high concentrations of applied pesticides (e.g., chlorpyrifos-methyl, oxidazon) were detected at playgrounds located near orchards and vineyards [86]. Up to 36 of 500 analyzed pesticide ingredients were detected in ambient air at each of 69 sites all over Germany using passive air samplers, even in national parks in mountainous forests [87]. Pendimethalin, prosulfocarb and eleven other semi-volatile substances were found in more than half of passive air samplers. The authors conclude that medium- and long-range transport likely accounts for these findings.

### 4.3. Enhanced Degradation Rates

Higher temperatures usually lead to a more rapid breakdown of organic chemicals in the environment through microbial conversion and/or through abiotic processes such as hydrolysis and photolysis, leading to a decrease in their total lifetime in the environment. Lamon et al. predicted this for various POPs in the northern hemisphere by applying a complex atmospheric–oceanic circulation model, assuming an increase in mean global temperature of 3.4 °C by the year 2100 [88]. The authors further forecasted that climate change will enhance the long-range transport of PCBs and other POPs from the regions of their former use in the northern hemisphere to neighboring continents and to the arctic regions. The faster breakdown of organic chemicals at higher temperatures can be advantageous because the persistence is then less pronounced. However, predictions are difficult to make: potentially toxic conversion products may also be formed more quickly and in higher concentrations. The consequences resulting from climate change, such as drought and reduced vegetation, may entail that the pollution of soils, bodies of water and living organisms does not ultimately decrease despite elevated temperatures.

### 4.4. Increased Toxicity and Ecotoxicity

Rising temperatures often lead to an increase in human and ecotoxic effects of pollutants in the environment [78]. It is well-known from both eco- and human toxicology that the action of two or more stressors on a living being is usually more harmful than if only one of the stressors acts with the same intensity. This applies regardless of whether the stressors are chemical, physical or biological or a combination. Physical and biological stressors can be of natural origin, such as unfavorable temperature or water and food shortages or pathogenic organisms. Climate change can exacerbate these natural stressors or cause them in the first place. Organisms that cannot escape to other habitats are threatened with extinction. For example, it has been proven that the temperature tolerance of fish decreases when they are exposed to active ingredients of various pesticides. The combined effects of high temperature and a chemical can lead to a harmful effect, even if the individual doses on their own have no noticeable effect. For example, Besson et al. [89] found that combined exposure to a non-toxic concentration of the insecticide chlorpyrifos and an increased water temperature of only 1.5 °C disrupts the thyroid hormone system in fish. Chemical and non-chemical stressors that occur together in aquatic organisms can lead to a considerable increase in the toxic effect, by the authors' denominated "over-additive" effect [90].

Toxicants can impair the ability of mammals and birds to adapt to changing environmental conditions. For example, the concentrations of polychlorinated biphenyls (PCBs) and per- and polyfluorinated alkyl substances (PFAS) in the bodies of some populations of polar bears are so high that they weaken the body's defenses [91]. The animals' immune system is then less able to withstand infections and to adapt to climate stress. For the adaptability of animals to higher temperatures, it is not only the absolute increase but also the speed of global warming that is decisive [92]. Therefore, the combination of the over-average fast warming of the arctic regions and the high POP exposure of top predators must be regarded as a threat to the biodiversity of arctic wildlife.

*4.5. Heavy Rainfall and Floods Mobilize Pollutants*

As one consequence of global warming, it is expected that the frequency and intensity of heavy precipitation and floods will increase. If—as is often the case—municipal sewage treatment plants can no longer cope with substantial amounts of rain, untreated wastewater diluted with rainwater will directly flow into streams and rivers [93]. Accordingly, higher amounts of pollutants from wastewater will find their way into surface waters.

In the event of floods, pollutants in contaminated sediments can be remobilized. As an example, from 1930 to 1945 large quantities of magnesium were produced in the region north of Leipzig. The conversion of the raw materials with chlorine gas resulted in the unintentional formation of polychlorinated dibenzodioxins and -furans (PCDD/F) as by-products, which were discharged with the washing water into the rivers Mulde, Bode and Saale. As a result, over the past 70 years floods have mobilized these substances and contaminated the sediments and floodplains of the Elbe up to the Port of Hamburg and the confluence with the North Sea [79]. In the same region, the GDR produced large amounts of organochlorine pesticides in the catchment area of the river Mulde, which accumulated in its sediment. During the major flood event in August 2002, these pollutants were remobilized and washed into the river Elbe. In the following years, significantly higher levels of these pesticides were detected in breams. (Figure 4) [94].

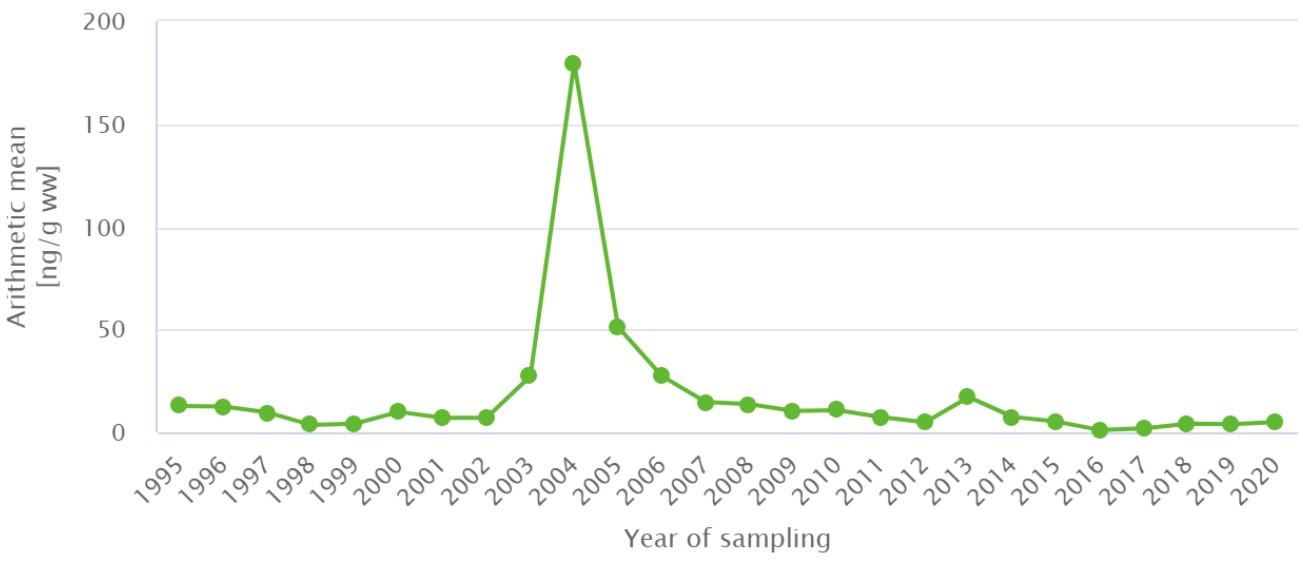

**Figure 4.** Concentrations of beta-hexachlorocyclohexane in muscles of bream (Source: Environmental Specimen Bank, Umweltbundesamt [94]).

## 5. Extraction, Production and Use of Substances Endanger Biodiversity

According to the IPBES (Intergovenrnmental Platform on Biodiversity and Ecosystem Services) report in 2019, biodiversity is decreasing dramatically around the world. The natural extinction rate, which up to around AD 1500 was 0.1 to 2.0 species per million species and year, is now exceeded by a factor of ten to one hundred times. Around a quarter of the known species of vertebrates, invertebrates and plants are threatened with extinction [95]. However, the decline in biodiversity is not limited to the number of species that are becoming extinct: the total population of wild vertebrates has decreased by 68% since 1970 [96]. When ecosystems are exposed to stress, it is the sensitive species that disappear first. Even though tolerance of ecosystems against anthropogenic pollution,

so-called "pollution-induced community tolerance" (PICT), often increases [97], the ecosystems are at risk anyway. The ability of an ecosystem to return to its original stable state after disruptions—the so-called "resilience"—decreases as diversity declines [98].

In the Global Biodiversity Outlook 5 (GBO 5), it is demonstrated that most of the 20 Aichi targets—targets which were stipulated at the Conference of the parties of the Convention on Biological Diversity in 2010—will not be achieved [99,100]. The main reasons for this are changes in land use due to agriculture and forestry, especially deforestation, as well as urbanization and overexploitation of ecosystems such as the oceans due to fishing. Additionally, climate change and invasive species contribute to the loss of species. Furthermore, the extraction of raw materials, the production and use of materials, the use of pesticides [101] and fertilizers endanger biodiversity. In the opinion of the authors of GBO 5, reversing the progressive loss of biological diversity also requires lower pollution, sustainable production and reduced levels of consumption.

### 5.1. Raw Materials Extraction and Processing

The continuous increase in material flows begins with the extraction of the raw materials. Extraction of ores and fossil raw materials increased dramatically in the past century and this continues in the 21st century [102,103]. As an important driver a further doubling of consumption of resources is expected by 2050 (Figure 5) [104]. On the other hand, overall raw material productivity is increasing, e.g., in Germany by 35% between 2000 and 2016 [105]. However, these gains were more than offset by the sharp increase in extraction of non-renewable resources triggered by the so-called rebound effect. Since extraction from ever lower concentrations requires steadily increasing effort, exhaustion of mineral resources is within reach. If the coveted metals are only accessible in low concentration ores, even larger spoil heaps arise during extraction and processing. In the case of open-pit mines in particular space requirements and the resulting loss of life-supporting landscape are enormous. If ore content is decreasing, mining waste and the use of chemicals and consumption of energy are increasing [106]. In the Global South, the extraction of raw materials is also associated with violations of human rights in a number of cases.

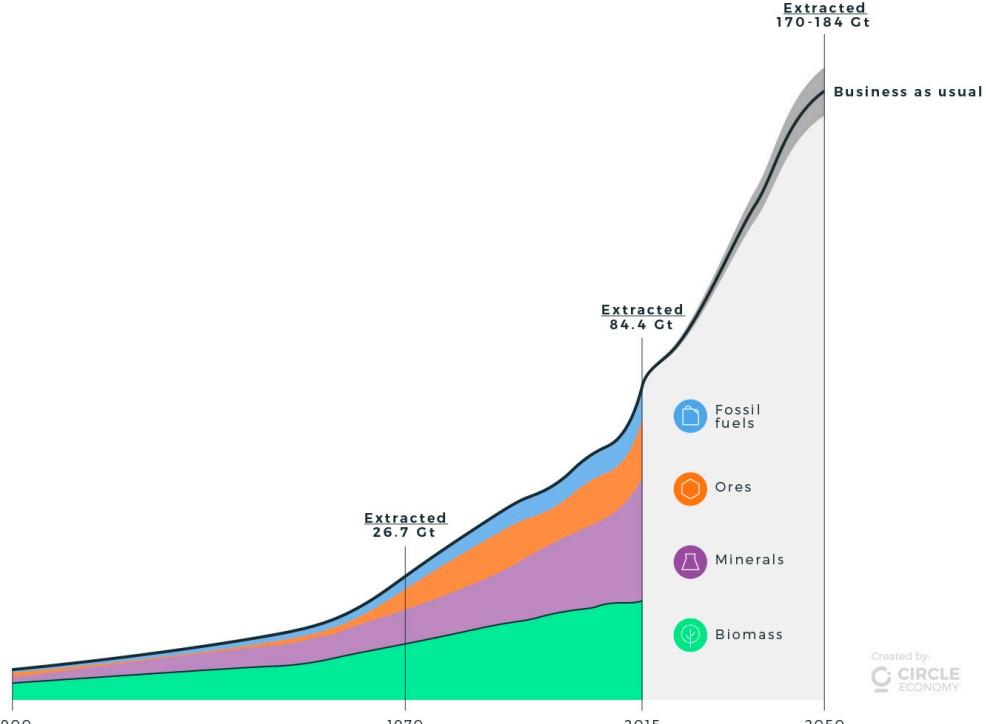

**Figure 5.** Worldwide increase in raw material extraction (according to de Wit et al. [104]); Biomass, Minerals, Ores, Fossil Fuels; graphic: Kay van 't Hof, Circle Economy.

Often, toxic by-products are released into the environment during mining activities, e.g., the use of mercury by small-scale gold prospectors in the Amazon Forest or the release of uranium and cadmium when phosphate is extracted. Dramatic examples of natural habitats being destroyed by the extraction of raw materials are the extraction of oil from tar sands in Alberta (Canada) and the contamination of the Niger Delta in Africa with mineral oil. Not only single species are threatened but entire ecosystems are also destroyed.

The processing of ores also leads to substantial toxic emissions and thus endangers thriving natural habitats. From the 19th and 20th centuries, flue gas from smelter processing plants is known to have caused the death of forests in Germany and also the loss of mammals and birds in the vicinity [107,108]. Thanks to improved cleaning of the exhaust gas in such systems, this has become a past event in the EU but is still widespread in the countries of the South. In the more recent past, after dam breaches in Hungary, Canada and Brazil, toxic ore processing sludge has exacted a toll of many lives and poisoned entire valleys and rivers. It should also be kept in mind that, for a long time, the sintering of iron ore also was an important source of emissions for polychlorinated dibenzodioxins and -furans (PCDD/F) [109,110].

*5.2. Toxic Emissions (Air, Water, Waste)*

When organic substances are burned, not only are carbon dioxide and water produced, but also numerous other substances that pollute the environment and the biosphere and are then distributed across vast areas. For a long time, the emissions of sulfur dioxide ($SO_2$) and nitrogen oxides (NO and $NO_2$) from the burning of fossil fuels numbered among the most pressing environmental problems in industrialized countries. Sulfuric acid evolves from $SO_2$ emissions, resulting in low pH values in rainwater, particularly damaging forests. Nitrogen oxides—especially their emissions from diesel engines are still a problem today—are oxidized to nitric acid and, in addition to acidification, cause extensive fertilization and thus eutrophication of terrestrial ecosystems. Incomplete combustion may lead to the emission of toxic polycyclic aromatic hydrocarbons (PAHs).

When incinerating waste and especially products containing chlorine in any form, polychlorinated dibenzodioxins and -furans (PCDD/F) can be generated. These are among the most toxic substances are persistent and bioaccumulate in food chains. That is why they are regulated in the Stockholm Convention as Persistent Organic Pollutants (POPs). In industrialized countries, these emissions are now being minimized using the best available techniques (BAT). Thus, sustainable chemicals and materials management affords the avoidance of emissions from incineration, but also from the manufacture and processing of materials and substances, especially of toxic substances. If this is not possible, it is necessary to reduce them to a level as low as possible. In many cases, harmful emissions can be avoided by choosing less hazardous substances and changing production processes, respectively. In addition, precautionary measures must be taken to ensure that operational disruptions, which can lead to the unintentional discharge of harmful substances, are avoided.

Whereas BAT have been required by EU Regulation for a long time, one toxic substance in the exhaust air from coal-fired power plants was not strictly regulated in Europe: mercury, a poisonous heavy metal which, among other aspects, causes loss of intellectual abilities in children [111]. Mercury is contained in small amounts in coal and is only incompletely filtered out from flue gases. The Minamata Convention calls for measures to reduce these emissions, which are higher than any other entry route of mercury into the global environment. The easiest way to avoid such emissions is obvious: to no longer generate energy from burning coal.

In industrial processes and also when chemicals are used in households and businesses, wastewater is generated that often contains hazardous substances. Eighty percent of global wastewater ends up in water bodies, mostly in rivers, without being treated [87]. Numerous substances in sewage cannot, or can only partially, be filtered out by sewage treatment plants. One example is the numerous active ingredients in medicinal products that mainly

end up in water through human excretion [112]. Purified wastewater effluents still contain numerous chemicals, mostly in low concentrations; but, when assessing the effects on aquatic communities, it must also be taken into account that combined effects of toxicants may harm the aquatic biocenosis. For some chemicals, environmental quality standards (EQS) for surface waters are stipulated in the EU or at the national level [113]. Toxic substances also make their way into the marine environment via rivers. A reduction in water pollution from wastewater ingredients can be achieved by substitution of problematic pollutants, especially persistent pollutants, in products and processes, by implementing measures at the source, and further by expanding sewage treatment plants to include a fourth purifying stage.

Further pollution is caused by household and commercial waste: people dispose of substances and products that they no longer use by discarding them with household waste. If these items cannot be reused or recycled, they must be disposed of by landfill or incineration. The EU's framework directive on waste also regulates the handling of hazardous waste [114] which can pose a risk to the biosphere. Despite EU waste legislation and the international Basel Convention, in countries of the Global South, the (illegal) shipments of significant amounts of electrical and electronic scrap containing dangerous substances such as heavy metals and brominated flame retardants are a huge problem [115]. There, some valuable metals are recovered under conditions that are harmful to health and the environment [116].

*5.3. Plastic Threatens the Biosphere*

A particular threat for the biosphere is the dissemination of plastic. Plastic is mostly non-toxic, but extremely persistent. Once released into the environment, it is difficult to retrieve, especially if it has been added to products as microplastics or if it spreads into the environment as abrasion from plastic items, textiles or vehicle tires. Especially in countries without a functioning waste management system, plastic waste ends up in the environment and ultimately ends up in the oceans mainly via rivers. Marine organisms, especially plankton feeders, ingest the particles, are unable to digest them and starve to death although their stomachs are full. Microplastics can also enter higher organisms such as fish and marine mammals as well as the human body via the food chains [117]. The pollution of the oceans with microplastics and macroplastics poses an existential threat to the marine environment. The irreversibility of marine plastic pollution and its global dimension induced researchers of the Stockholm Resilience Center to evaluate whether plastic alone exceeds the planetary boundaries for novel entities [118]. They concluded that, although there is still ignorance regarding disruptive effects on the marine environment, precaution is needed and means to stop further pollution should be taken urgently. Soils (via compost, digestate and sewage sludge), rivers and lakes also contain ubiquitous plastic residues [119]. Plastic is everywhere. The large quantities in combination with their scarce degradability make plastics one of the most pressing global environmental problems today. In addition, toxic additives in plastics, such as plasticizers, antioxidants, UV stabilizers and flame retardants, often pose a serious environmental problem [40]. Plastic particles in the environment can also adsorb pollutants from the aqueous environment, accumulate such pollutants and transport them into the bodies of organisms.

Until recently, plastic waste was exported from the EU on a large scale for recycling, mainly to Asian countries. The recycling there is often inadequate and leads to considerable environmental pollution [45]. However, some of these countries—especially China—no longer allow imports from Europe and North America, which leads to higher amounts of waste that is incinerated. The resolutions of the Basel Convention of 10 May 2019 [120] that in future only pure, unpolluted plastic waste may be exported for recycling without a permit is welcomed and to be consistently implemented. What is needed is an effective recycling system in the industrialized countries (see above).

*5.4. Pesticides and Fertilizers*

"The real emissions of the chemicals industry . . . are . . . the products themselves." This quote from Eberhard Weise, a former management board member at Bayer AG, demonstrates that products which are deliberately released into the environment, can pollute it and also threaten biodiversity [121]. This applies in particular to agricultural pesticides, biocides and fertilizers.

Agriculture is one of the most important drivers of the decline in species [122]. The cause is the form of industrial cultivation to which, for example, unploughed strips, where a lot of birds and small mammals live, fall victim. Modern agriculture is also based on the intensive and widespread use of pesticides and fertilizers. In particular, the insecticidal neonicotinoids damage biological diversity in and around the fields. Compared to 1995, the amount of pesticide active ingredients applied in agriculture has not increased in Germany—it is approximately 30,000 tons annually—but today's active ingredients are many times more effective than they were then [122]. It should also be borne in mind that the areas where pesticides are applied to are generally decreasing due to the rise of organic cultivation [123].

However, it is not just a matter of the direct toxic effects on target organisms. Indirect effects are also decisive: for example, herbicides such as glyphosate eliminate wild herbs which the pollinator insects then lack as food [101,124]. Food chains are broken; animals starve to death at what appears to be a richly set table. The situation is exacerbated by the cultivation of herbicide-resistant crops, which means that the quantities used per acre increase, crop rotation is avoided, and resistant wild herbs develop [125].

As declared at the 8th Global Nitrogen Conference (INI), the natural nitrogen cycle has now doubled in scale due to human activities, driven by intensive animal agriculture, over-fertilization of agricultural land and fossil fuel combustion. Nitrogen's planetary boundary is estimated to have been substantially exceeded by humanity. The unique chemistry of the nitrogen cascade means that it exacerbates a range of environmental and human-health problems central to sustainable development, from air pollution and biodiversity loss in terrestrial and aquatic ecosystems, to climate change and pollution of drinking water [126].

Fertilizers also reduce agrobiodiversity [127]. In particular, the nitrogen content of the soils especially in countries with extensive livestock breeding is exceeding threshold concentrations as a result of the application of liquid manure and mineral fertilizers. This also applies to grassland which is only used to a small extent for grazing. Due to intensive fertilization, the diversity of the different types of grassland is decreasing [128]. Nutrient-rich and species-poor meadows are becoming the norm. Above all, plants with high nutrient requirements such as nettles and goutweed are spreading. The accompanying flora, which was important for the arable habitat, disappears. This is a major cause for the dramatic decline in flying insects and, as a result, of birds in agro-ecosystems [129]. Mammals such as the brown hare or hamster and birds such as the gray partridge, Eurasian skylark or yellow hammer are among the animal species threatened with extinction. The disappearance of many animals, plants and microorganisms from agro-ecosystems leads to a reduction in ecosystem performance, and thus endangers the long-term stability of agricultural landscapes. Additionally, the diversity of soil life is severely impaired by intensive cultivation and the use of chemical products [130]. Industrialized agriculture also leads to soil leaching. The content of bound organic carbon, among other forms as humus, decreases and the soil degrades.

It is becoming increasingly evident that pesticides also affect bacterial communities (microbiomes). In particular the common herbicide glyphosate (trade name: Roundup) damages the intestinal microflora of mammals and pollinating insects such as bees by blocking the shikimate pathway [131–134]: the composition of bacterial flora changes, immunity to pathogens decreases, and the reproduction and behavior of the host animals changes. The microbial diversity in the soil is also adversely affected by pesticides (herbicides, fungicides, insecticides) [135]. Soil fertility and the diversity of higher soil organisms decrease.

Intensive agriculture not only reduces the diversity of plant and animal species in agricultural areas. The variety of cultivated plants is also steadily decreasing. A few high-performance cultivars among cultivated plants are becoming increasingly dominant on arable land. Farmers usually cannot obtain their own seed from hybrid seeds but have to buy new seeds every year. The agro-ecological adaptation to local and regional cultivation conditions is decreasing. The high-performance varieties usually require a high level of nutrients, are sensitive to harmful organisms and susceptible to plant diseases. This leads to intensive use of fertilizers and pesticides.

### 5.5. Endocrine Disruptors and Infochemicals

Hormonally active substances have a particular mode of action in organisms. Endocrine disruptor (ED) substances influence the hormonal systems of humans and animals even at low concentrations. They simulate or block hormones (especially the sex hormones and the thyroid hormone) or change formation, transport and metabolism of natural hormones [136]. Numerous industrial chemicals and pesticides act this way. Endocrine disruptors that act like sex hormones can change the hormone balance of embryos in early pregnancy and thus contribute to disturbances in the development of the genital organs. Further, they may even contribute to the development of breast or prostate cancer. Hormones prescribed as medication are also emitted into the environment through excretion [112]. Endocrine disruptors not only threaten human health but can also damage natural communities in ecosystems. For example, the gender ratio of fish in polluted waters has changed in favor of female animals. Reproduction and development of both vertebrates and invertebrates are influenced [137].

Hormones are messenger substances within an organism. In natural systems, information is also exchanged between living creatures through substances, which are called infochemicals. Organisms such as insects and plants use chemical substances to inform each other about food sources, predators, etc. Insects send out sexual attractants (pheromones). Other chemicals influence the swarming behavior of fish and grasshoppers [138,139]. Infochemicals play a decisive role in vital processes such as reproduction, social behavior, eating, defense or orientation in most living beings. Some fragrances in cosmetics or detergents and cleansing agents have structures that are very similar to those of natural infochemicals [140]. Initial findings show that some synthetic chemicals can simulate or suppress such information [141]. Effects like these are even observed from substances that are present in plastic waste [142,143]. As a result, anthropogenic infochemicals are potential disruptive factors in a highly sensitive system of coexistence between animal and plant species [144–146]. For example, some neuropharmaceuticals impair the swarming behavior of fish while some pesticides impair the sense of direction of honeybees.

### 5.6. Cultivation of Renewable Raw Materials

Replacing fossil raw materials with renewable raw materials appears to be the ideal solution to sustainability, seemingly overcoming the climate crisis and reducing material risks for biodiversity. In the recent past, demand for biomass from nature has increased significantly [87]. Biological raw materials are an important approach for a climate-neutral economy. However, this approach has clear limits (see above). Renewable raw materials can only replace fossil raw materials to a very limited extent, in any case. Wood—a desirable alternative to mineral construction materials such as concrete in the construction sector—is also a limited resource, especially since, from the point of view of nature conservation, more uncultivated, near-natural forests are desirable to protect biodiversity. The increasing competition for land is worrying: if more and more energy crops are grown, they displace cultivation areas for food-delivering crops [147]. Some energy crops such as *Silphie* or *Miscanthus* are increasingly grown in Europe but do not yet have evolutionary established accompanying flora. Thus, their cultivation may lead to further species impoverishment. Renewable raw materials are imported from the countries of the Global South to a greater and greater extent. However, as the case of palm oil shows, a great deal of natural habitats

is destroyed in order to obtain these renewable raw materials. The clearing of primeval forests not only in tropic zones results in serious ecological damage, such as leaching of the soil, damage to the water balance, and also leads to the release of greenhouse gases as a result of the degradation of the soil, so that the climate balance is negative. However, in Europe too, increasing cultivation of biomass has consequences: fallow land, which was an important retreat for endangered animal species and plants, has been taken back under the plow. While around 10% of agricultural land in Germany was taken out of use at the end of the 1990s, it is only 2% to 3% today [122].

*5.7. Invasive Species and Global Trade*

Invasive species are an important driver of the threats posed to global biodiversity. These are organisms that are carried off or imported from other parts of the world, colonize new habitats and can displace the native species there (Neobiota) [148]. Very often, it is the increasing international transportation of goods that causes this spread. Many organisms such as termites or spiders travel on merchant ships as "stowaways": wood from China brought the Asian longhorn beetle to Germany [149]; together with their load or their tires, trucks overcome natural barriers such as the Alps and bring the Asian tiger mosquito to Central Europe. The ballast water of ocean-going ships is full of alien organisms and so it must be disinfected for international shipping in accordance with the Ballast Water Convention. This international agreement was adopted in 2004 and has been in force since 2017 [150]. If the organisms that have been introduced find favorable conditions at the destination, they can establish themselves there and multiply. The increase in global flows of chemicals and materials is favorable to the spread of alien animals and plants. Among the organisms that are spread in this way are mosquitoes and other organisms that can transmit zoonoses (diseases transmitted by organisms).

*5.8. Reduced Chemical Diversity*

Impacts of substances threaten biological diversity. Conversely, the decline in species also creates problems: ecosystem services that are important for production, such as pollination, are at risk. Furthermore, many genetic resources, the protection of which is regulated by international treaty, the Nagoya Protocol 2010 [151], are lost, and with them interesting biologically active substances. This means an irretrievable loss of chemical diversity. Nature already offers a large number of drugs, including antibiotics. However, numerous plant ingredients with a pharmacological effect have not yet been discovered. The complex, species-rich ecosystem of the coral reefs and the tropical rainforests contain innumerable, as yet unknown substances and messenger substances that can be important for pharmaceuticals, crop protection or technical applications. These genetic resources need to be preserved in order not to lose options for suitable, environmentally compatible solutions.

## 6. Elements of a Sustainable Chemicals and Materials Management

How can chemistry contribute to sustainability? On the one hand, chemistry provides many tools which help to achieve the Sustainable Development Goals (SDGs). We need chemicals for renewable energy generation and storage, clean water, hygiene and health, mobility or corrosion protection. On the other hand, chemicals may pose a threat to man and environment and may compromise the SDGs. Above all, sustainable chemistry implies at least to avoid irreversible harm to human health and ecosystems [152,153]. This can be achieved by

- Chemical products which do not have hazardous characteristics that burden the environment and health;
- Chemical production carried out in such a way that it does not involve any danger for human beings or the environment and efficient in regard to energy and resources;
- Regeneration and recycling which are taken into account from the very beginning;
- Material flows managed in such a way that they do not exceed planetary boundaries and satisfy ecological criteria.

Sustainable chemistry involves both the scientific discipline of chemistry and also chemistry as an economic sector. It includes the twelve principles of "green chemistry" of Anastas and Warner as scientific criteria [154] and in addition examines the functions chemical substances should carry out. Thus, it also includes social, economic and ethical aspects. The International Collaboration Center for Sustainable Chemistry (ISC3) has listed the central characteristics of sustainable chemistry [155].

Many functions can be carried out without the use of synthetic chemicals. However, a "chemistry-free" world is no longer imaginable and would also be unsustainable.

### 6.1. Sustainable Chemicals

Chemicals to which humans or the environment can be exposed should, if possible, have no hazardous properties. In particular, they should not be persistent, should not accumulate in the long-term and should not trigger any as yet undetected, irreversible effects. They should be "benign by design" [156], that is, structurally benign, and have a short range in terms of time and space ("short range chemicals") [157]. Some hazard characteristics are indispensably linked to their intended function. For example, by their nature fuels have to be combustible and disinfectants must be toxic to target organisms. However, hazardous properties that do not fulfill a desired function should be avoided whenever possible.

Further criteria such as the resource and energy demand during extraction and manufacture, the suitability for application, the possibility of reuse and utilization (recycling) or composting, and the behavior in the waste phase play an important role when assessing the sustainability of chemicals. Germany's Federal Environmental Agency has published a decision-making aid as a "Guide on Sustainable Chemicals" [158]. UNEP recently published a Framework Manual on Green and Sustainable Chemistry [159]. Moreover, the sustainability of the function a chemical is expected to fulfill can also be decisive [160,161].

The chemical-leasing business model is an example of how the application of chemicals can be reduced, thus avoiding hazards [162]. Traditionally, manufacturers are interested in selling their product to their customers in bulk or in large numbers. This inflates the flow of materials more than is necessary. However, if a supplier offers a service such as cleaned workpieces, lubricated equipment or hygiene in addition to the substance, it is also in their interest to use as little material as possible. This creates a win–win situation that has economic benefits for both sides. There are still obstacles that hinder general adoption of this approach: some industry partners fear know-how losses or too great a dependence on a single supplier. In some cases, liability issues have not been clarified. There is urgent need to develop solutions to overcome these barriers. New business concepts are also needed to avoid waste, reuse products or to recycle them as valuable products [163]. At present, it is still easier and cheaper for many traders to "dispose of" waste, which is why a circular economy is still a distant goal [164].

### 6.2. Sustainable Production of Chemicals

Chemicals should be manufactured in a way that meets sustainability criteria. This means, among other aspects, high energy and resource efficiency, effective purification of sewage and exhaust air, low waste generation and inherently safe production processes. This requires innovative changes to current production methods: for example, changing the raw material base, bio-technical and (new) catalytic processes, use of microreactors, and conversion to syntheses at low temperature and low pressure [165].

### 6.3. Sustainable Materials Flow Management

Facing the steady increase in production, use and disposal of substances, strategies are needed to slow down and reduce material flows. The volume of chemical production is currently duplicated every 12 years; in particular, short-lived products increase in consumption which results in rising transport and waste volumes. Furthermore, the plurality of chemicals used in products increases. If this kind of management is continued, we will

overshoot planetary boundaries and risk a collapse of Earth system and economy. A fundamental change is necessary. Following the aim of sustainable material flows, three strategic approaches, which complement each other, should be followed: efficiency, consistency, and sufficiency.

The efficiency strategy is based on the saving and long-term use of materials. Considerable progress is made to promote efficiency in production and processing of resources. This means specific resource consumption decreases in relation to value creation. However, this decoupling of resource consumption from GDP development can be challenged by increased consumption. In many areas, consumption is contributing to an acceleration of materials flows. Savings in expensive resources can even generate a "rebound effect", when the saved money is spent for a higher consumption rate [166].

The aim of the consistency strategy is to integrate material flows as well as production and consumption patterns into natural cycles and ecosystems as far as possible. This relates especially to the sustainable use of natural (bio)resources (input into the technosphere) as well as the assimilation capacity of natural systems regarding emissions and waste (output into ecosphere, just another quantifiable (global or regional) boundary). Within the technosphere, the circular economy is in focus. This is realized, e.g., by constructions that consist of modules or materials that can be reused unchanged in other places that are durable, repairable and can be used several times. Another aspect is the recovery of (secondary) raw materials and use of high value recycling products. In addition to recycling and resource-efficient production, a consistent economy relies on the aims of a sustainable bioeconomy regarding the interaction between the biosphere and technosphere and within the technosphere. It is based on the use of materials and technologies that are of biological origin, that are based on natural models or that are constructive or functional copies of natural structures (biomimetics). However, there are limits to this strategy, as is also the case with the efficiency principle: the cradle-to-cradle approach assumes that a complete technical return or transfer of products and materials into technical or biological cycles is possible [167]. Nature is successful in managing material cycles, especially bio–geo–chemical cycles, in a more or less closed manner. However, in technical practice dissipation poses limits to recycling. In addition, hazardous as well as persistent and mobile non-natural substances pose limits to the assimilation capacity of the Earth system.

The sufficiency strategy poses the question of what is enough. In concrete terms, this means consciously minimizing the consumption of resources. If this strategy is translated as "frugality", then it is frowned upon in consumer societies because it sounds like shortages and austerity [168]. However, sufficiency does not mean austere sacrifice, which raises the question of the right amount and a more conscious handling of limited resources [169,170].

Steps toward significantly higher efficiency and consistency can be implemented in a socially and economically acceptable manner and are largely undisputed. Corresponding optimizations of value chains or business models prove this in practice. For greater acceptance and a breakthrough for the sufficiency principle, growing awareness of the finiteness of resources, new guiding ideas for prosperity and growth, a suitable legal framework with a policy offering incentives, and thus a departure from the economic-growth model, are required. However, we will not achieve the goal of sustainable materials management without effective sufficiency strategies.

The concept of the circular economy focuses primarily on cycles within the technosphere. The aim is to ensure that products made from chemical substances have the longest possible product lifetimes. They should be suitable for reuse, extended use, repair and high-quality recycling. Regional material flows should consistently have priority over global material flows. In addition, the question ultimately being considered is whether the cycles of the techno- and ecosphere are open or closed to one another. Humankind extracts and uses biomass and other raw materials from the ecosphere, but this should be managed in a sustainable fashion. On the other side, in the case of easily degradable, compostable materials, it is possible to open up technical cycles to global biogeochemical cycles and regional ecological cycles. Persistent substances, and toxins or nutrients that

disrupt natural material cycles and cannot be assimilated in nature, need to be sealed off from the ecosphere (Figure 6) [171].

### Opening and Closing Material Cycles

**Figure 6.** There are two different cycles with options of flow regulation, closing or opening. The cycle within technosphere is currently named as 'circular economy'. Another cycle exists between ecosphere and technosphere. It can be interpreted as a kind of metabolism: resource consumption, digestion/processing, excretion/waste/emissions (adapted from Gößling-Reisemann et al. [171]).

Particular attention should be paid to all chemicals such as fertilizers, pesticides, pharmaceuticals or substances for coating surfaces that humans deliberately introduce into the environment in order to achieve certain effects. Here, the question of the benefit, the minimization and the non-chemical alternatives must be posed even more clearly. Complete and rapid mineralization is then a prerequisite for sustainable use.

The greatest challenge of the 21st century is to achieve a sustainable and globally tolerable and just way of life and a balanced economy. A sustainable chemicals and materials management can and must make a significant contribution to achieving the climate and biodiversity goals. What is needed is a socio-ecological restructuring of our society, a transformation oriented toward common welfare taking into consideration the limitations posed by the planetary boundaries. A reorientation of resource and materials management is a central component of this transformation. The aim must be to change the quality and significantly reduce the quantity of materials and substance flows. Thus, sustainable material flow management requires not only sustainable chemistry, but also an ecological economy [172]. The report "The Economics of Biodiversity" of Dasgupta [173] and the corresponding policy guide of OECD [174] present ways of giving economic value to nature and thus achieving the Sustainable Development Goal of sustainable production and consumption. According to this report, fundamental changes in the economic and financial system, but also in institutions and social values, are necessary in order to bring nature back into equilibrium with human activities.

### 7. Conclusions

Sustainable chemicals and materials management must focus intensely on the persistence and mobility of chemicals and on the flow of materials from raw material extraction over processing and production to reuse and disposal. Therefore, chemicals and materials management must be based on the following guiding principles:

Chemicals and materials management is international today. The pollution of the Earth system with "novel entities" has reached a critical level. In order to counteract this,

the United Nations Sustainable Development Goals must be taken seriously and binding measures must be taken to implement them.

Material flows must be slowed down and reduced both regionally and globally. Above all, this means using fewer non-sustainable chemicals. Moreover, greater resource efficiency, consistency and sufficiency in the handling of substances and materials are necessary strategies towards sustainable material cycles.

Chemicals and materials management and circular economy must be linked. A reduction in material flows can only succeed if the waste hierarchy is systematically considered.

Sustainable chemicals and materials management is everybody's business: (i) the state, which influences the behavior of companies and consumers through concrete regulations and authorizations, as well as through indirect incentives such as taxes and levies, with the aim of fostering a sustainable economy and safe handling of substances; (ii) the companies that live up to their responsibilities in a global economy and provide sustainable products, as well as (iii) consumers who, on the basis of correct and specific information, orient their lifestyle to the principles of sustainability.

The most important principles of the sustainable management of substances were already described and illustrated in 1993 (Figure 7) [175]: following these, "ecological design" is the central issue. This design aims at using only chemicals and substances whose environmental impact is as low as possible. Thus, only substances should be used that are as non-persistent as possible but are also neither mobile nor toxic and also do not accumulate in the food-web. To reduce the overall exposure of humans and ecosystems, the consumption of chemicals should be reduced by increased efficiency, consistency and sufficiency.

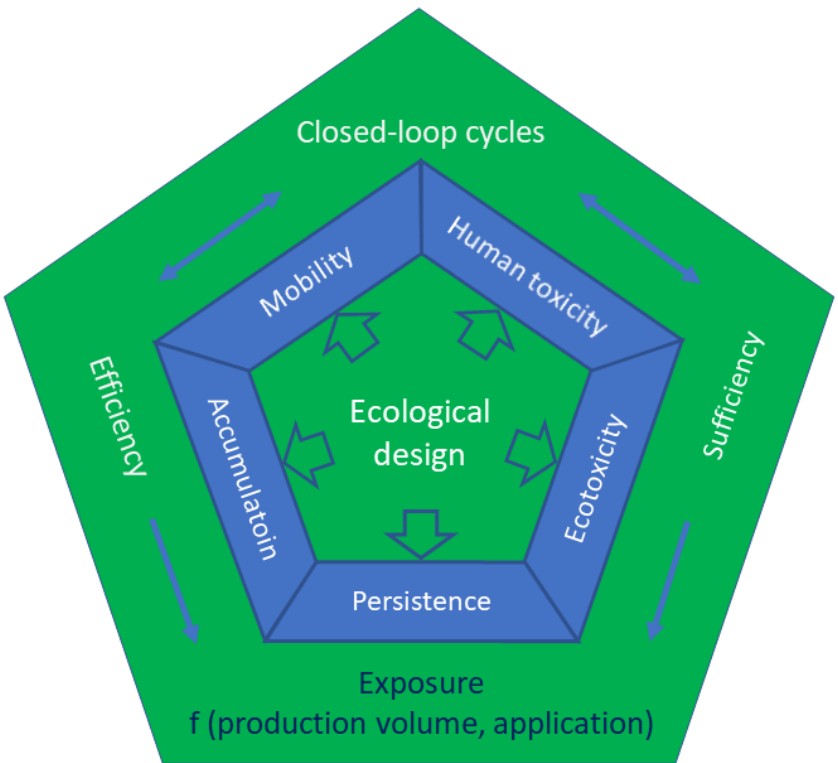

**Figure 7.** Minimizing the risks of chemicals by reducing exposure and effects (adapted from Friege [175]).

The loss of biological diversity, climate change and the exceeding of the boundaries of the Earth system by an inadequate quality and quantity of substance and material streams are closely intertwined. All three global challenges must be tackled and solved together. It is becoming more and more evident that global environmental problems are the result of an economy that is based on growth of production and consumption. There is

an urgent need to turn away from the constantly increasing consumption of energy and resources. This will not be achieved on the required scale simply by decoupling economic growth and resource consumption. Rather what is needed is an approach that includes consistent sufficiency strategies, especially in the countries of the Global North, turning away from perpetual material growth. The countries of the Global South need a sustainable development strategy that enables economic participation without emulating the waste of energy and resources as practiced by the Global North.

The three approaches must interlock: first, efficiency strategies, then consistency strategies, that is, in particular the recycling of materials and the use of materials and technologies that can be integrated into ecological cycles, as well as sufficiency strategies that avoid excessive use of resources.

Dramatic climatic changes are not the only consequences of excessive energy and resource consumption, but also the spread of hazardous substances and the increasing depletion of non-renewable resources. The global threats ultimately lead to a loss of natural spaces and biodiversity and, associated with this, pose a threat to human life and its survival.

Both at the international level and in the European and national framework, the interactions between biodiversity, climate and chemicals and materials policy must be taken into account and addressed more closely. An international process must be initiated at all levels, which, based on the sustainability goals of the United Nations, defines a common framework for action that links all three policy areas and sets out specific action goals and instruments analogous to Agenda 21 at the World Conference in Rio de Janeiro in 1992. Therefore, we recommend developing an international framework convention on sustainable chemicals and materials management. The serious global environmental changes show: time is of the essence. Fast, networked and consistent action is required.

**Author Contributions:** All authors contributed to the entire text of this review. K.G.S. drafted Sections 3 and 5 and contributed to Sections 6 and 7. A.V.G. focused on the 'sustainable materials flow management' (Section 6.3) and the introduction (Section 1). M.G.O. reviewed the global aspects in Section 2, contributed to the text on sustainable chemistry (Section 6) and drew some conclusions in Section 7. W.K. provided the contents of Section 4. All authors have read and agreed to the published version of the manuscript.

**Funding:** This research received no external funding.

**Data Availability Statement:** Data presented in this review are publicly available and can be found under the references listed below.

**Acknowledgments:** We thank the working group "Environmental chemicals and toxicology" of FoE Germany (BUND), in particular Henning Friege, Dieter Cohors-Fresenborg, Janna Kuhlmann, Manuel Fernandez, Volker Molthan and Ralph Ahrens for helpful and critical discussions and excellent proposals.

**Conflicts of Interest:** The authors declare no conflict of interest.

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
