# Peer review of "The Necessity of a Global Binding Framework for Sustainable Management of Chemicals and Materials—Interactions with Climate and Biodiversity"

_2673-4079, doi:10.3390/suschem3020014_

Round 1

Reviewer 1 Report

It is hard to give detailed review comments on the article as it needs significant editing to tell a clearer "story".  The abstract is exceptionally long and rambles with many unconnected ideas.  The article is simply too long.  If the thesis is "A global chemicals and materials framework convention integrating the current fragmented and non-binding approaches is urgently needed." then the article should talk about that as the starting point and then why this is the case - eg chemicals contribute to climate, biodiversity, and are exceeding the safe planetary boundary.  There is lots of fine detail in the article but even within sections material jumps from issue to issue.  

The authors need an editor to take all of this good research, give it a clear thesis or purpose - and then shape the narrative and information to support that purpose.  Otherwise, the article is a string of very good and important but unconnected ideas and the reader is left unclear of what the goal fo the piece is.  

Author Response

Thanks for the critical comments. We now tell a clearer story and put the thesis that we need a framework convention at the beginning in the introduction. This thesis is now no longer part of the conclusion. We also changed the title accordingly. The abstract is now significantly shorter. Considering the positive comments of reviewers 2 to 5 we did not implement more fundamental changes.

Reviewer 2 Report

After carreful reviewing, it can be said that this paper corresponds to the aim and scope of the Journal. The manuscript looks solid and well-organized, and meets typical requirements to review papers. Actually, I have no major critical remarks towards this paper; however, it is recommended to replace figure 2 with more recent data.

Author Response

Thanks for the comment. Much to our regret, we could not find a more recent figure 2 with updated data, neither in the publications of IPCC nor at the UNEP website and other sources. To our knowledge, there was not much change in the last 5 years. The only alternative would be to remove the figure.

Reviewer 3 Report

The review „Sustainable Management of Chemicals and Materials – Implications to Climate and Biodiversity“ of authors Klaus Günter Steinhäuser, Arnim von Gleich, Markus Große Ophoff and Wolfgang Körner deals with multiplex problems joined with actual unsustainable consumption of our civilization. This extensive review is divided into 5 main chapters (except Introduction and Conclusion) and many subchapters discussing drawbacks of current management of materials based on significant utilization of non-renewable resources with low degree of recycling. The authors discuss some new areas of waste treatments enabling re-use of spent materials according to the circular economy principles.

This review is appropriate for publishing in Sustainable Chemistry journal, the reviewed area of interest should be very attractive for the readers of this journal.

Please, add the Reference to the statement mentioned on pages 3-4, lines 116-117.

Page 4, lines 163-164: „…(e.g., PFOS and PFOA1)…“, please, replace the superscript with explanation of these abbreviations.

Page 5, line 186: Please, replace „auf“ in: „3.1. Expenditure of energy for production and use auf substance“.

Page 7, line 262: Please, replace the superscript with explanation of this term „…the more exergy2“…

Page 9, lines 377-378: Could you add the Reference for this statement, please? By my best of knowledge, the lignin derivatives dissolved in aqueous solution are used as the biofuel during paper and cellulose production. Using lignin and its derivatives as the source of chemicals, the other sources of energy must have been used in this biorefinery-based area of industry.

Page 10, Subchapter 3.4: Could you develop discussion dealing with PtX (utilization of CO2 together with H2) involving potential of another more available reactive chemicals suitable for chemisorption  and material utilization of CO2 (epoxides, etc.)? For more details, see for example: Weidlich T., Kamenicka B.: Utilization of CO2-Available Organocatalysts for Reactions with Industrially Important Epoxides. Catalysts, 2022, 12(3), Article No.: 298. DOI: 10.3390/catal12030298 or Artz, J.; Müller, T.E.; Thenert, K. Sustainable Conversion of Carbon Dioxide: An Integrated Review of Catalysis and Life Cycle Assessment. Chem. Rev. 2018, 118, 434−504. https://doi.org/10.1021/acs.chemrev.7b00435.

Page 13, Figure 4: The description in Figure 4 should be translate from German language to English for reader´s better understanding.

Page 14, line 566: Please, explain the abbreviation “IPBES“.

Page 14, line 567: Please, explain the abbreviation “AD”.

Page 24, Figure 7: Has the letter “f” in this Figure beside “production volume, application” some relevance?

Author Response

We thank the reviewer for detailed and helpful editing. With regard to his comments

  • Pages 3-4, lines 1116-117: We added the reference 12 (Steffen et al.)
  • Page 4, PFOS and PFOA: Unfortunately, the footnotes disappeared in the manuscript. The explanation by footnotes is now added at the end of the text before the references.
  • Page 5, “auf”: corrected. Is changed to “of”
  • Page 7: the term exergy is explained by a footnote (see above)
  • Page 9 (lignin): Thanks to reviewer for this information. Text revised
  • Page 10 (Utilization of CO2): Thanks for the interesting references that are now added. However the text is modified 1 page earlier, when the “carbon capture and use (CCU)” is mentioned.
  • Page 13 Fig 4: English figure included
  • Page 14: IPBES explained by a footnote
  • Page 14: AD means “Anno Domini”. If necessary we can replace it by “in the year (1500)”
  • Page 24, figure 7: f means “function of”. Exposure is determined by production volume and kind of application.

Reviewer 4 Report

Comments

Klaus Günter Steinhäuser et al. presented a very interesting work on the topic of Sustainable Management of Chemicals and Materials. The result of this work will be useful for integrating a global chemicals and materials framework convention. The discussion and conclusion of this work were supported by various survey. Therefore, this work can be considered for publication in Journal of Sustain. Chem..

(1) Can the explanation of figure 1 be more sufficient and clear?

(2) Please explain the meaning of some nouns where they appear for the first time, such as‘novel entities’.

(3) Please check whether the phrase 'contribute to'' is appropriate for the title of the part 3.

(4) In part 3.1, What is auf substances?

(5) ‘Thus, a shift of energy production away from fossil fuels would also help to reduce GHG emissions caused by the production of chemicals and materials.’ What about the alternative?

(6) Please check the format of the full text..

(7) Whether part 3.5 is not sufficient and reasonable?

(8) Does the title need to be changed to show the interaction between chemicals and materials on climate and biodiversity?

Author Response

Thanks for the comments.

  • The explanation of figure 1 is now clearer
  • Some footnotes added. The term novel entities is explained in the first paragraph after figure 1.
  • “contribute to” replaced by “interact with”.
  • “auf” replaced by “of”
  • The alternative are renewable resources. Text amended
  • We do not understand what the reviewer means with “format of the full text”
  • Part 3.5: We do not think that the topic needs more text and is explained sufficiently clearly. Transport of materials and goods needs energy independently from energy expenditure of production.
  • Title of the manuscript was changed

Reviewer 5 Report

The manuscript is a comprehensive review of  issues and policies related to sustainable chemicals and materials management as related to climate and biodiversity. The manuscript is well written and addresses the many issues involved in a clear way. My only two comments are that the abstract is quite long. It should be condensed to about half its current size, and that in the introduction a clear statement about the objectives of the paper should be included.

Author Response

Thanks for the comment. Introduction and Abstract have been changed. The statement is now clearer. The Abstract is now significantly shorter (35-40%).